# Leukodystrophy Imaging: Insights for Diagnostic Dilemmas

**DOI:** 10.3390/medsci12010007

**Published:** 2024-01-25

**Authors:** Rajvi N. Thakkar, Drashti Patel, Ivelina P. Kioutchoukova, Raja Al-Bahou, Pranith Reddy, Devon T. Foster, Brandon Lucke-Wold

**Affiliations:** 1College of Medicine, University of Florida, Gainesville, FL 32610, USA; 2College of Medicine, Florida Atlantic University, Boca Raton, FL 33431, USA; 3College of Medicine, Florida International University, Miami, FL 33199, USA; 4Department of Neurosurgery, University of Florida, 1600 SW Archer Rd., Gainesville, FL 32610, USA

**Keywords:** leukodystrophies, demyelinating disorders, magnetic resonance imaging, myelin imaging, white matter disorders

## Abstract

Leukodystrophies, a group of rare demyelinating disorders, mainly affect the CNS. Clinical presentation of different types of leukodystrophies can be nonspecific, and thus, imaging techniques like MRI can be used for a more definitive diagnosis. These diseases are characterized as cerebral lesions with characteristic demyelinating patterns which can be used as differentiating tools. In this review, we talk about these MRI study findings for each leukodystrophy, associated genetics, blood work that can help in differentiation, emerging diagnostics, and a follow-up imaging strategy. The leukodystrophies discussed in this paper include X-linked adrenoleukodystrophy, metachromatic leukodystrophy, Krabbe’s disease, Pelizaeus–Merzbacher disease, Alexander’s disease, Canavan disease, and Aicardi–Goutières Syndrome.

## 1. Introduction

Leukodystrophies, a group of rare and debilitating disorders affecting the white matter of the central nervous system, present a complex interplay between pathology and genetics. This literature review explores the intricate landscape of leukodystrophies, delving into the pathological mechanisms that underlie these disorders, the genetic intricacies that drive their manifestation, and the imaging techniques used as part of the diagnosis regimen. Through a synthesis of recent research findings, this review aims to provide a comprehensive understanding of the histopathological alterations within the central nervous system, shedding light on how these changes contribute to the clinical presentation and progression of leukodystrophies.

Simultaneously, this review delves into the pivotal role of genetics in leukodystrophies, where monogenic mutations significantly impact the disease onset, severity, and progression. From advancements in genetic testing methodologies to the identification of novel genes associated with specific leukodystrophy subtypes, the genetic landscape of these disorders is rapidly expanding. This synthesis of the latest research in pathology, genetics, and imaging aims to offer a robust resource for clinicians, researchers, and genetic counselors striving for a deeper understanding of leukodystrophies and the development of targeted therapeutic interventions.

Magnetic resonance imaging (MRI) is the primary imaging technique to identify, localize, and characterize cerebral lesions in patients with leukodystrophy. These disorders pose a threat to the integrity of the brain and peripheral nerves, with clinical presentations often being nonspecific [1,2]. Imaging techniques, particularly MRI, play a crucial role in establishing a definitive diagnosis [3]. Current research suggests that early detection of leukodystrophy allows for more optimal implementation of therapy treatments, highlighting the importance of early disease detection [3,4].

Quantitative MRI, providing insights into myelin and axonal content, condition, and white matter composition, aids not only in diagnosis, but also in understanding disease progression [5]. Additionally, genetic testing, focusing on distinct alterations in specific genes, complements the diagnostic process, with advanced sequencing significantly accelerating the speed of diagnosis [6,7,8].

This review specifically concentrates on elucidating the most common leukodystrophies that are classified: X-linked adrenoleukodystrophy, metachromatic leukodystrophy, Krabbe’s disease, Pelizaeus–Merzbacher disease, Alexander’s disease, Canavan disease, and Aicardi–Goutières Syndrome, offering a nuanced exploration of their clinical presentations and distinctions seen on MRI to facilitate conclusive and specific diagnoses. (Figure 1).

## 2. Types of Leukodystrophies

### 2.1. X-Linked Adrenoleukodystrophy

X-linked adrenoleukodystrophy (X-ALD), an X-linked genetic condition impacting the central and peripheral nervous systems along with the adrenal cortex, primarily manifests in boys and clinically presents with adrenal insufficiency, dysarthria, dysgraphia, vision, and hearing deficits, as well as neurocognitive and neurobehavioral issues [9,10]. The condition is marked by a mutation in the *ABCD1* gene on the X chromosome, which is responsible for encoding the ALD protein, a transmembrane protein crucial for the transport of very long chain fatty acid (VLCFA)-CoA esters into the peroxisome [11,12]. Consequently, mutations in the *ABCD1* gene lead to diminished VLCFA transport and widespread accumulation throughout the body [13]. Adrenocortical insufficiency typically develops in nearly all affected males around age 7.5, with the onset of progressive myelopathy and peripheral neuropathy occurring in adulthood [9,11]. Sixty percent of male patients will experience progressive and often fatal cerebral demyelination, a phenomenon detectable via the instrumental use of MRI, facilitating early detection and improving patient outcomes [14,15].

Newborn screening for X-ALD has gained widespread recommendation for implementation into the uniform screening panel in the United States, with over half of the states initiating screening. The diagnosis of X-ALD requires the detection of the *ABCD1* pathogenic variant and the accumulation of VLCFAs [9]. Cerebral X-ALD cases present with distinctive white matter demyelination and inflammation, visualized on MRI, with lesions categorized into three zones [11,16,17]. These zones delineate the loss of axons, oligodendrocytes, and myelin sheaths, highlighting gadolinium enhancement and active macrophage involvement. Brain MRIs exhibit hyperintense confluent lesions of the corpus callosum and parieto-occipital white matter, progressing to impact the entire cerebral white matter in adulthood [18,19,20,21,22]. Routine brain MRIs are conducted to monitor disease progression, contrasting with the less utilized spinal cord imaging, which demonstrates corticospinal tract and dorsal column degeneration [23,24,25]. Furthermore, spinal cord MRI demonstrates corticospinal tract and dorsal column degeneration, resulting in the appearance of a flattened spinal cord and the reduction in anteroposterior diameters [23,24,25]. The pathophysiology of X-ALD, succinctly captured in Figure 2, underlines the prevention of VLCFA entry into the peroxisome due to *ABCD1* gene mutations, resulting in the clinical phenotype arising from VLCFA accumulation throughout the body.

While MRI remains the gold standard for lesion detection in X-ALD, emerging studies in other neurodegenerative disorders suggest the potential utility of Neurite Orientation Dispersion and Density Imaging (NODDI) in revealing increased orientation dispersion and a higher surface area of neurodegeneration compared to structural MRI and diffusion tensor imaging [26,27]. Additionally, myelin water fraction (MWF) imaging, assessing the quantity of myelin via specific water pools, presents a promising avenue for detecting early X-ALD lesions in the future [28].

### 2.2. Metachromatic Leukodystrophy

Metachromatic leukodystrophy, an autosomal recessive lysosomal storage disease, is characterized by a deficiency of arylsulfatase A (ARSA) due to a mutation in the *arylsulfatase A* gene on chromosome 22q13.3-qter [29,30]. ARSA plays a crucial role in the degradation of sulphatide, a membrane lipid found in myelin, the distal tubules of the kidney, and bile duct epithelia [29]. The deficiency of ARSA leads to the accumulation of sulphatide primarily affecting the nervous system and resulting in progressive demyelination, presenting clinically with ataxia, optic atrophy, dementia, and decerebrate posturing [29,31].

Children suspected of metachromatic leukodystrophy often exhibit delays in meeting developmental milestones and a decline in both gross and fine motor skills [30]. Diagnosis involves laboratory studies to assess ARSA levels, with criteria ranging from undetectable to less than 10% of the normal value [10]. Distinguishing metachromatic leukodystrophy from arylsulfatase A pseudodeficiency, present in about 1% of the general population, requires additional assessments such as urine sulfatide levels, radiolabeled sulfatide fibroblast loading, and DNA analysis.

As a demyelinating condition, MRI reveals brain demyelination and abnormalities in nerve conduction [32,33]. Initial impact occurs in the central and periventricular white matter, progressing to subcortical structures. Extreme cases may exhibit projection fiber involvement, leading to distinctive patterns like the “tigroid pattern” and “leopard skin” pattern as shown in Figure 3 [34,35]. Nearly all patients with metachromatic leukodystrophy show splenial corpus callosum demyelination [36]. T2-weighted FLAIR images display symmetric, confluent hyperintense areas in the periventricular white matter, consistent with the demyelinating nature of the disorder [37]. Lastly, MRI modalities, including diffusion-weighted parameters, demyelination load, and MR spectroscopy, hold potential in aiding early diagnosis by providing insights into nonspecific white matter changes associated with metachromatic leukodystrophy [38,39,40].

### 2.3. Krabbe’s Disease

Krabbe disease (KD), an inherited lysosomal storage disease, poses a diagnostic challenge due to its diverse clinical presentation and overlap with other neurodegenerative disorders [41]. This disease, also known as Globoid Cell Leukodystrophy, results from a defect in the *GALC* gene, leading to the accumulation of toxic myelin products [42]. KD typically manifests in infants under the age of six but can also occur in adolescents or adults, presenting symptoms such as muscle weakness, spasticity, hypertonia, myoclonic seizures, and sensory deficits [42,43]. At the presymptomatic stage, stem cell transplant can improve patient prognosis; however, without a family history of KD, the disorder is often not identified until the symptomatic stage. Once symptomatic, the patient becomes ineligible for transplantation [44]. For this reason, the goal of care is to screen infants for KD and identify it before symptoms appear.

Infant screening for KD is limited across the U.S., highlighting the importance of early identification before symptom onset [45]. Consensus guidelines recommend a three-step screening, diagnosis, and treatment process, which is detailed in Figure 4 [44]. The initial step involves a dried blood spot assay for *GALC* enzyme activity, followed by diagnostic tests at a specialty care center (SCC). Neurodiagnostic studies, including MRI and CSF analysis, are conducted at a human stem cell transplantation center (HSCT) upon confirmation of the diagnosis.

The MRI findings in KD exhibit distinctive patterns across different subtypes, such as optic nerve and cervical cord enlargement in the infantile form and T2-hyperintense changes in corticospinal tracts in the adult form [46,47,48]. Psychosine accumulation in the CSF serves as a diagnostic tool, offering potential insights into gene therapy targets [47,49,50]. Despite its severity, KD diagnosis relies on a combination of clinical, genetic, and imaging assessments, emphasizing the significance of early screening and multidisciplinary approaches for optimal patient management.

### 2.4. Pelizaeus–Merzbacher Disease

Pelizaeus–Merzbacher disease (PMD) stands out as a rare leukodystrophy and CNS demyelinating disease, with its clinical manifestation documented by Friederich Pelizaeus and Ludwig Merzbacher in 1885 [51]. This neurological disorder presents symptoms such as nystagmus, spastic paralysis, and ataxia, leading to a progressive decline in coordination, motor skills, and cognitive function [52]. The X-linked recessive inheritance of PMD is attributed to mutations in the proteolipid protein 1 (*PLP1*) gene on the X chromosome, resulting in varied levels of decreased myelin production [51]. Classification into Types I, II, and III is contingent on the specific mutation, with Type I being the most severe [51].

Diagnosing PMD poses complexity due to overlapping symptoms with other leukodystrophies, necessitating the exclusion of alternative diagnoses [51]. Neonatal onset presents a more severe prognosis compared to the nearly benign adult form [53]. Inoue et al. introduced an interphase fluorescence in situ hybridization (FISH) assay for efficient screening, successfully diagnosing PMD and detecting carriers [54]. Molecular analysis further aids in identifying the size and location of gene duplications. Imaging via MRIs reveals gliosis around demyelinated areas, with T1-weighted sections indicating hypointensity and T2-weighted images displaying hyperintensity, reflecting demyelination (Figure 5). Magnetic resonance spectroscopy (MRS) findings in PMD cases vary, necessitating additional research for conclusive diagnostic utility [55,56]. Sumida et al.’s retrospective study correlated MRI scans with disease severity, emphasizing high T2-weighted intensity in the brainstem or corticospin tract [57,58]. CSF analysis indicates a low level of N-acetyl aspartate (NAA), correlating with axonal damage and disease severity [57,58].

While FISH, MRI, and CSF analysis contribute to PMD identification, the reliance on exclusionary diagnosis due to overlapping findings underscores the need for further research. Current treatments focus on supportive care to alleviate symptoms, emphasizing the imperative for ongoing research to enhance the diagnosis and treatment of PMD and ultimately improve patients’ quality of life.

### 2.5. Alexander Disease

First identified in 1949, Alexander disease (AxD) constitutes a form of leukodystrophy affecting the CNS white matter, marked by myelin sheath degeneration due to a defect in the Glial Fibrillary Acidic Protein (*GFAP*) gene [59,60]. Although predominantly associated with infants, clinical manifestations can arise at neonatal, infantile, juvenile, and adult stages [61]. A defining diagnostic feature of AxD is the presence of Rosenthal fibers and eosinophilic granular bodies observed via light microscopy and categorized into two subtypes: Type I and Type II [62].

The exclusive defect linked to AxD originates from mutations in the *GFAP* gene on chromosome 17q21.31, responsible for encoding a protein expressed in astrocytes. This mutation leads to the formation of *GFAP* aggregates, known as Rosenthal fibers, causing astrocyte degeneration and demyelination [63]. Quantification of *GFAP* levels, notably elevated in the cerebrospinal fluid (CSF) of AxD patients, provides valuable diagnostic insight, although marginal elevations may occur when measured in blood [64].

Regarding imaging, MRI observations typically reveal abnormal signals in the frontal white matter, periventricular rim, or structures such as the caudate head, thalamus, and brainstem [65]. Figure 6 illustrates increased T2-weighted signal intensity and decreased T1-weighted signal intensity on MRI, aligning with AxD characteristics [66]. Van der Knaap et al. outlined five MRI criteria, necessitating the fulfillment of four for an imaging-based diagnosis: extensive cerebral frontal white matter abnormalities, periventricular rim with altered T2 and T1 signals, abnormalities in basal ganglia and thalami, brainstem irregularities, and contrast enhancement in specific structures [65]. Notably, hindbrain structural abnormalities, including brainstem atrophy and cervical spinal cord signs, are distinctive features of later-onset presentations [60].

Lastly, while autopsies are the only definitive diagnostic tool for Rosenthal fiber analysis, current diagnosis techniques revolve around histological examination via brain biopsy, MRI diagnosis, and gene analysis for the *GFAP* gene [67].

### 2.6. Canavan Disease

Described for the first time in 1931, Canavan Disease stands as an autosomal recessive leukodystrophy predominantly impacting the brain’s white matter [68]. The ASPA gene on chromosome 17p13.2 is implicated, encoding aspartoacylase; a deficiency in this gene leads to the accumulation of N-acetyl-L-aspartic acid (NAA), associated with oligodendrocyte dysfunction and myelin degradation [68]. Primarily affecting infants, the disease is more prevalent in Ashkenazi Jews, although occurrences in other populations have been documented [69]. Excessive NAA is believed to disrupt myelination pathways, contributing to abnormal myelination and tissue spongy degeneration [69,70]. While there are no curative therapies, ongoing research explores the introduction of the ASPA gene into functional neural progenitor cells, showing promise in mouse models [71,72].

Confirmation of Canavan disease involves culturing skin fibroblasts with NAA and assessing NAA levels via gas chromatography-mass spectrometry after incubation; elevated levels indicate aspartoacylase deficiency, distinguishing it from other leukodystrophies like Alexander’s disease, which exhibit normal NAA levels. Aspartoacylase expression can also be determined via chorionic villus biopsy for prenatal diagnoses [73,74].

Similar to other white matter diseases, Canavan disease manifests hyperintense T2-weighted images in subcortical U fibers [75]. T1 signals appear hypointense with diffuse signals throughout the white matter and brainstem [76]. MRI reveals diffuse cerebral white matter degeneration, with preserved structures like the periventricular rim. Follow-up MRI scans for infants may exhibit progressive ventriculomegaly and atrophy [77,78]. Other modalities, such as CT, may depict white matter hypodensity [78].

### 2.7. Aicardi–Goutières Syndrome

Aicardi–Goutières Syndrome (AGS) stands as a rare genetically inherited neuroinflammatory disorder impacting the brain, immune system, and skin, presenting progressively with symptoms such as dystonia/spasticity, hepatosplenomegaly, elevated liver enzymes, thrombocytopenia, chilblain-like skin lesions, and neurological abnormalities, including microcephaly, CSF lymphocytosis, and developmental delays [79,80,81]. These symptoms, reminiscent of TORCH congenital infections despite the absence of active viral infection, have led to the term “Pseudo-TORCH syndrome” for AGS. The diagnosis involves a multifaceted approach, encompassing a comprehensive understanding of its distinctive clinical presentation, specific imaging findings, intricate genetic underpinnings, and discerning cerebrospinal fluid (CSF) and bloodwork biomarkers [82,83,84]. Effective management and prognosis evaluation often necessitate vigilant follow-up imaging procedures to monitor disease progression and assess treatment efficacy.

MRI study findings play a pivotal role in identifying characteristic brain abnormalities associated with AGS. These findings commonly include the loss of white matter, particularly in the periventricular and deep white matter regions, as well as calcifications in the basal ganglia and dentate nuclei [79,81,85]. Additionally, the presence of ventriculomegaly and the thinning of the corpus callosum are frequently observed, contributing to the distinctive radiological profile that aids in distinguishing AGS from other neurological conditions [80].

MRI studies play a crucial role in identifying characteristic brain abnormalities associated with AGS, including loss of white matter in periventricular and deep regions, calcifications in the basal ganglia and dentate nuclei, ventriculomegaly, and the thinning of the corpus callosum [79,81,85]. AGS, inherited in an autosomal recessive pattern, is closely linked to genetic mutations involving genes associated with the intracellular metabolism of nucleic acids, such as TREX1, RNASEH2A, RNASEH2B, RNASEH2C, SAMHD1, ADAR, and IFIH1 [79,82,86,87,88,89,90]. These mutations lead to increased calcium deposits in the brain, believed to result in an overactive immune system [91]. The strong association between AGS and a predisposition to other autoimmune conditions, like systemic lupus erythematosus, has been revealed in recent research [92,93].

Specific abnormalities in the CSF, such as elevated levels of interferon-alpha, neopterin, and other proinflammatory cytokines, along with the presence of autoantibodies, aid in differentiating AGS from related conditions. Advanced genomic sequencing methods, including Next-Generation Sequencing (NGS) and Whole-Exome Sequencing (WES), have enhanced diagnostic accuracy and facilitate the identification of novel genetic variants. Follow-up imaging, particularly serial MRI examinations, is crucial for monitoring disease progression, tracking the evolution of abnormalities, and optimizing individualized management strategies, underscoring the significance of a multidisciplinary approach in AGS care.

## 3. Conclusions

In conclusion, the spectrum of leukodystrophies presents a complex and diverse array of disorders, each with distinct pathological and genetic underpinnings. This comprehensive review has delved into the intricacies of various leukodystrophies, exploring their pathology, genetic manifestations, diagnostic modalities, and implications for patient management.

Leukodystrophies, whether X-linked adrenoleukodystrophy, Metachromatic leukodystrophy, Krabbe’s disease, Pelizaeus–Merzbacher disease, Canavan disease, or Aicardi–Goutières Syndrome, share commonalities in their impact on the white matter of the central nervous system. Through advanced imaging techniques like magnetic resonance imaging (MRI), clinicians can discern specific patterns of demyelination and other structural alterations critical for accurate diagnosis and prognosis. Moreover, the integration of advanced genomic sequencing methods, such as Next-Generation Sequencing (NGS) and Whole-Exome Sequencing (WES), has significantly improved diagnostic precision, allowing for the identification of novel genetic variants and a deeper understanding of disease mechanisms.

While significant progress has been made in elucidating the genetic landscape and diagnostic tools for leukodystrophies, challenges persist. Diagnosing these disorders remains intricate due to overlapping clinical presentations, necessitating a meticulous exclusion of other leukodystrophies. In the absence of a definitive cure for many leukodystrophies, supportive and symptomatic management remains the mainstay of treatment. The importance of early detection, especially in disorders like X-linked adrenoleukodystrophy, highlights the potential impact on therapeutic interventions and patient outcomes.

As we continue to unravel the complexities of leukodystrophies, ongoing research endeavors, innovative diagnostic technologies, and collaborative multidisciplinary approaches offer hope for improved diagnostic accuracy, targeted therapeutic interventions, and enhanced quality of life for individuals affected by these challenging disorders. The integration of advanced imaging, genetic testing, and evolving diagnostic techniques marks a promising path forward in our pursuit of comprehensive understanding and effective management of leukodystrophies.

## Figures and Tables

**Figure 1 medsci-12-00007-f001:**
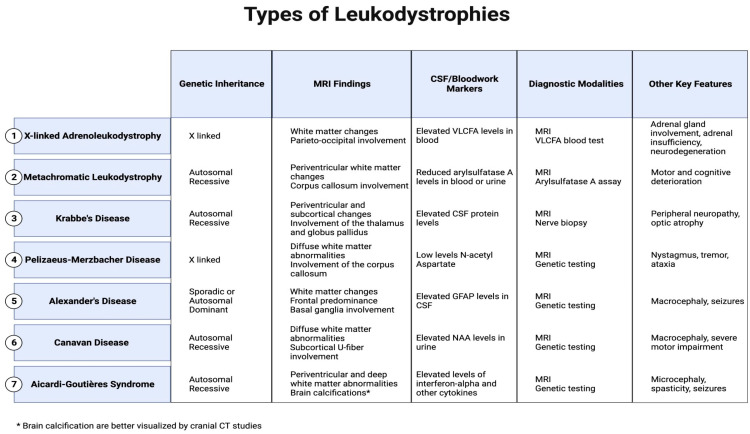
Major MRI findings, CSF/blood markers, and other key features in leukodystrophies.

**Figure 2 medsci-12-00007-f002:**
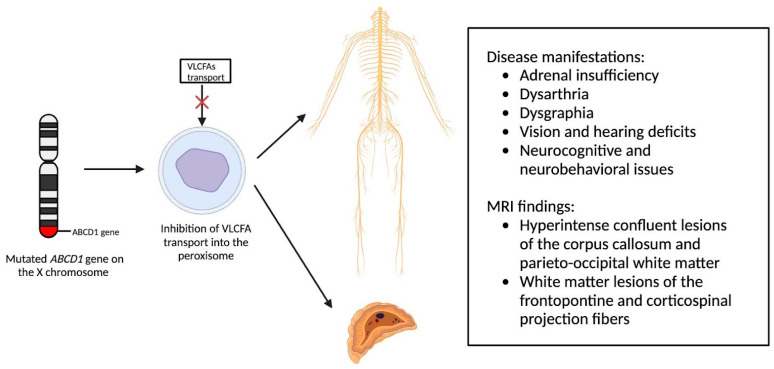
Pathophysiology of X-Linked Adrenoleukodystrophy: Mutations in the *ABCD1* gene on the X chromosome result in the prevention of very long chain fatty acids (VLCFAs) into the peroxisome, resulting in the clinical phenotype due to VLCFA accumulation throughout the body.

**Figure 3 medsci-12-00007-f003:**
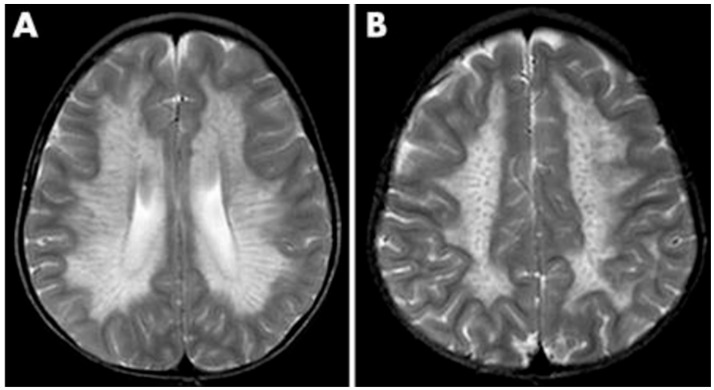
MRI findings found in metachromatic leukodystrophy. (**A**) Hypointense radial stripes resembling tiger skin. (**B**) Hypointense dots resembling leopard skin. Reprinted/adapted with permission from Ref. [35].

**Figure 4 medsci-12-00007-f004:**
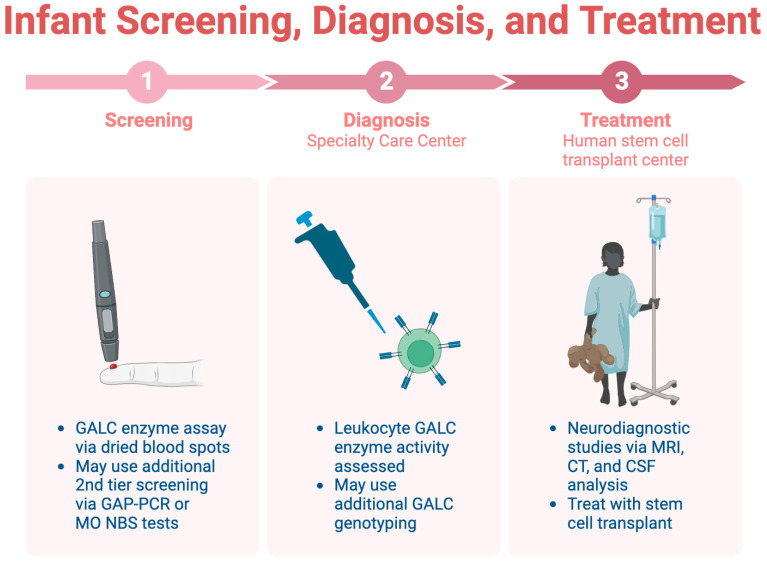
The three-step process for infant screening for KD.

**Figure 5 medsci-12-00007-f005:**
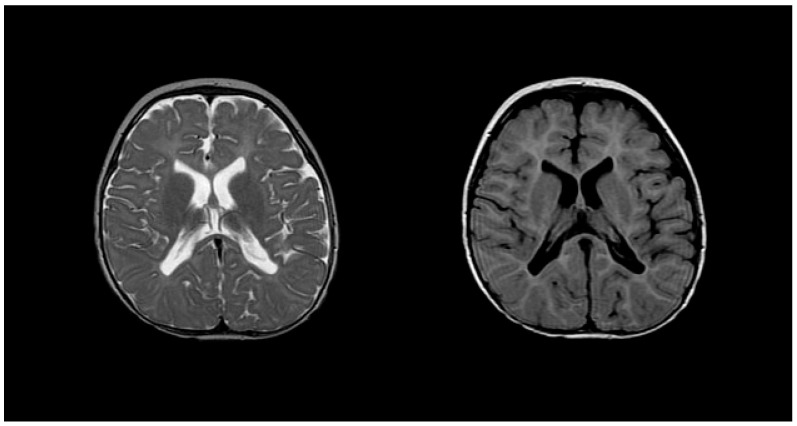
T2 (**left**) and T1 (**right**) imaging of a patient with Pelizaeus–Merzbacher disease showing lack of myelination in internal capsule, proximal corona radiata, and the optic radiation. Reprinted/adapted with permission from Ref. [57].

**Figure 6 medsci-12-00007-f006:**
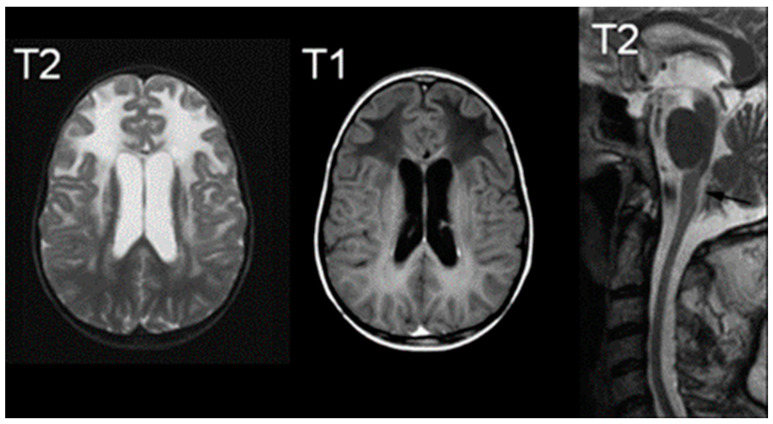
MRIs characteristic of Alexander Disease; the right image demonstrates an increased signal of T2, the middle image demonstrates a decreased signal of T1, and the right image is a T2 midline sagittal section demonstrating atrophy. Reprinted/adapted with permission from Ref. [66].

## Data Availability

No new data were collected or analyzed in this study. Data sharing is not applicable to this article.

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
