# Peer review of "Leukodystrophy Imaging: Insights for Diagnostic Dilemmas"

_medsci, 2024, doi:10.3390/medsci12010007_

Round 1

Reviewer 1 Report

Comments and Suggestions for Authors

This is an interesting and important review. However, I have some comments and suggestions which could improve the quality of this manuscript.

Introduction

  • Provides a broad overview of leukodystrophies and MRI diagnosis but lacks significant detail on pathology or genetics. Expanded background would strengthen it.
  • Writing style is repetitive and overly reliant on citing sources mid-sentence, disrupting flow. Varied syntax would improve readability.
  • Rationale for focusing on the 7 highlighted leukodystrophies is not provided. Some justification would help frame the scope.
  • Final statement that MRI assists diagnosis due to clinical overlap requires more support. Evidence of MRI's differentiation ability needed.

Individual Conditions

  • Sections provide good overviews of genetics, pathology, diagnosis, and imaging findings for each disease. However, sections follow a very similar structure throughout, making the writing repetitive.
  • MRI findings for each disease are described but better integration with pathogenesis explanations would enhance the synthesis.
  • Lacks critical analysis of limitations, uncertainties, or areas needing more research for diagnosing each disease. More balanced analysis would strengthen it.

Conclusion

  • Summary of using MRI to differentiate leukodystrophies based on patterns is clear but quite brief for a conclusion. Expanding this section would improve it.
  • Discussion of other diagnostic methods beyond imaging is limited. More comparison of techniques would better highlight MRI's role.

Overall Critique

  • Provides a broad survey of topics but lacks depth and critical analysis. More detail and balanced discussion throughout would strengthen it.
  • Writing style is stoic and repetitive. Use of more creative, expressive language and varied syntax would improve the narrative.
  • Conclusion is too short and abrupt given the length of the sections before it. A more synthesized conclusion that pulls together key themes would enhance it.
  • Figures are useful but better integration with the main text would help readers connect the images to the descriptions.
  • Strength is the breadth covered, but more focus on critical analysis for future directions would elevate the manuscript as a whole.

Author Response

Introduction

Provides a broad overview of leukodystrophies and MRI diagnosis but lacks significant detail on pathology or genetics. An expanded background would strengthen it.

Thank you for pointing this out. I have added two paragraphs in the beginning that expand on this more.

The writing style is repetitive and overly reliant on citing sources mid-sentence, disrupting flow. Varied syntax would improve readability.

I have edited the paper and tried to remove the repetitive stuff. However, as all of them have similar pathophysiology, it was harder to change some things.

Rationale for focusing on the 7 highlighted leukodystrophies is not provided. Some justification would help frame the scope.

I have added a point that these 7 types are the most common ones, and thus, focused in this paper.

Final statement that MRI assists diagnosis due to clinical overlap requires more support. Evidence of MRI's differentiation ability needed.

Done.

Individual Conditions

Sections provide good overviews of genetics, pathology, diagnosis, and imaging findings for each disease. However, sections follow a very similar structure throughout, making the writing repetitive.

MRI findings for each disease are described but better integration with pathogenesis explanations would enhance the synthesis.

Lacks critical analysis of limitations, uncertainties, or areas needing more research for diagnosing each disease. More balanced analysis would strengthen it.

I’ve edited each section and hope everything flows better.

Conclusion

Summary of using MRI to differentiate leukodystrophies based on patterns is clear but quite brief for a conclusion. Expanding this section would improve it.

Discussion of other diagnostic methods beyond imaging is limited. More comparison of techniques would better highlight MRI's role.

The conclusion is expanded and a better summary of all leukodystrophies is provided.

Reviewer 2 Report

Comments and Suggestions for Authors

I thank the opportunity to review the manuscript entitled “Leukodystrophy Imaging: Insights for Diagnostic Dilemmas” sent for publication in Medical Sciences. The authors present an interesting idea about a review on leukodystrophies. There are, however, several recently published reviews with a similar presentation and discussion in this topic. Furthermore, this manuscript needs a new and original approach to make it more interesting for the specialist and more attractive for the general physician and researcher. I have several suggestions at this point for the authors: 

1. There are only two specific figures which were provided by the authors in their review. They are only related to Metachromatic Leukodystrophy and Alexander’s disease. In my opinion, there is a major need to include figures about other causes of inherited leukoencephalopathies, such as Krabbe’s disease and X-linked Adrenoleukodystrophy. As this represents a review manuscript about Leukodystrophies for the general neurologist and geneticists, it is essential to present examples with the neuroimaging features related to the main genetic forms.

2. It is necessary to put all the genes described in the text in italics (lines 57, 60, 72, 143, 168, 207, 246, 252, 276, 280, 331, 332). 

3. As there were several updates in the leukodystrophy classification in the last 15 years, since the highly important reviews by Schiffmann et al., 2009, Vanderver et al., 2015, and van der Knaap et al., 2017, my suggestion is to include in the introduction a more detailed description about the leukodystrophies. 

4. Figure 1 is quite confusing. It should be presented as a table in my opinion. There is an inappropriate column called “Imaging techniques for diagnosis”, but in the lines there is content about enzymatic assays, blood tests, and genetic testing. In the same Figure 1, another suggestion is to take out “adrenal gland involvement” from the MRI findings and include only as “Other key features”. In the same Figure 1, I suggest the inclusion of Alexander’s disease as a sporadic or autosomal dominant condition. Furthermore, in the last line about Aicardi-Goutières syndrome, I would include an asterisk to mention that brain calcifications are better visualized by cranial CT studies. 

5. Another suggestion is to include a picture or table briefly presenting the Loes score for X-linked adrenoleukodystrophy (X-ALD), including a picture with the hallmark neuroimaging features seen in X-ALD. 

6. In Figure 2, I suggest the correction of ABCD1 gene description to be properly described with italics. 

7. I suggest the use of the abbreviation “ARSA” for the arylsulfatase A. The gene should be presented in italics. 

Author Response

There are only two specific figures which were provided by the authors in their review. They are only related to Metachromatic Leukodystrophy and Alexander’s disease. In my opinion, there is a major need to include figures about other causes of inherited leukoencephalopathies, such as Krabbe’s disease and X-linked Adrenoleukodystrophy. As this represents a review manuscript about Leukodystrophies for the general neurologist and geneticists, it is essential to present examples with the neuroimaging features related to the main genetic forms.

I’ve added one more MRI finding. For the others, I’m waiting for original author’s permission. However, I also feel like adding too many images might convolute the paper. Though, if it might be helpful, I can seek out other sources too.

It is necessary to put all the genes described in the text in italics (lines 57, 60, 72, 143, 168, 207, 246, 252, 276, 280, 331, 332). 

Done

As there were several updates in the leukodystrophy classification in the last 15 years, since the highly important reviews by Schiffmann et al., 2009, Vanderver et al., 2015, and van der Knaap et al., 2017, my suggestion is to include in the introduction a more detailed description about the leukodystrophies. 

A more generalized introduction including this suggestion has been updated.

Figure 1 is quite confusing. It should be presented as a table in my opinion. There is an inappropriate column called “Imaging techniques for diagnosis”, but in the lines there is content about enzymatic assays, blood tests, and genetic testing. In the same Figure 1, another suggestion is to take out “adrenal gland involvement” from the MRI findings and include only as “Other key features”. In the same Figure 1, I suggest the inclusion of Alexander’s disease as a sporadic or autosomal dominant condition. Furthermore, in the last line about Aicardi-Goutières syndrome, I would include an asterisk to mention that brain calcifications are better visualized by cranial CT studies. 

Done

In Figure 2, I suggest the correction of ABCD1gene description to be properly described with italics. 

Done

I suggest the use of the abbreviation “ARSA” for the arylsulfatase A. The gene should be presented in italics.

Done

Thank you for the above suggestions. They were extremely helpful and I hope the edits made are makes the article more clear.

Round 2

Reviewer 1 Report

Comments and Suggestions for Authors

The author responded to my comments and suggestions very well. Thank you.

Author Response

Thank you!

Reviewer 2 Report

Comments and Suggestions for Authors

The authors have properly evaluated and reviewed their manuscript taking into account the suggestions and changes brought by the reviewer. I have some additional suggestions at this point: 

1. Figures 5 and 6 do not have a presentation as parts A), B), and C). Furthermore, Figure 5 does not disclose classical neuroimaging findings seen in Pelizaeus-Merzbacher disease as the axial section, for example, does not allow a more detailed look to several important brain areas, such as the basal ganglia. 

2. In the Abstract, in the first phrase, the authors describe "Leukodystrophies, also known as demyelinating diseases, mainly affect the CNS". This sentence must be restructured by the authors to bring it more sense and a proper meaning. Leukodystrophies do not have the same meaning to demyelinating diseases, which in fact describe other clinical conditions (as a group) including multiple sclerosis, neuromyelitis optica spectrum disorders (NMOSD) and MOGAD.  

3. There are some minor typos that should be corrected, such as "MRI imaging" (line 268) - MRI = "Magnetic resonance imaging" imaging - my suggestion would be MR imaging or MRI studies... 

Author Response

1. Figures 5 and 6 do not have a presentation as parts A), B), and C). Furthermore, Figure 5 does not disclose classical neuroimaging findings seen in Pelizaeus-Merzbacher disease as the axial section, for example, does not allow a more detailed look to several important brain areas, such as the basal ganglia.

I've added a better picture for Figure 5 which shows demyelination of the internal capsule. Furthermore, the figure summary (below the figure) explains which image is T2 vs T1, thus presenting an equivalent explanation to A), B), and C).

2. In the Abstract, in the first phrase, the authors describe "Leukodystrophies, also known as demyelinating diseases, mainly affect the CNS". This sentence must be restructured by the authors to bring it more sense and a proper meaning. Leukodystrophies do not have the same meaning to demyelinating diseases, which in fact describe other clinical conditions (as a group) including multiple sclerosis, neuromyelitis optica spectrum disorders (NMOSD) and MOGAD.  

This makes sense. I've changed the sentence to reflect this better.

3. There are some minor typos that should be corrected, such as "MRI imaging" (line 268) - MRI = "Magnetic resonance imaging" imaging - my suggestion would be MR imaging or MRI studies... 

This also makes sense. I've changed all the "MRI Imaging" to MRI studies.